# Identification of Clinical Variants beyond the Exome in Inborn Errors of Metabolism

**DOI:** 10.3390/ijms232112850

**Published:** 2022-10-25

**Authors:** Alejandro Soriano-Sexto, Diana Gallego, Fátima Leal, Natalia Castejón-Fernández, Rosa Navarrete, Patricia Alcaide, María L. Couce, Elena Martín-Hernández, Pilar Quijada-Fraile, Luis Peña-Quintana, Raquel Yahyaoui, Patricia Correcher, Magdalena Ugarte, Pilar Rodríguez-Pombo, Belén Pérez

**Affiliations:** 1Centro de Diagnóstico de Enfermedades Moleculares, Centro de Biología Molecular, Departamento de Biología Molecular, Universidad Autónoma de Madrid, Centro de Investigación Biomédica en Red de Enfermedades Raras (CIBERER), IdiPAZ, 28049 Madrid, Spain; 2Unit for the Diagnosis and Treatment of Congenital Metabolic Diseases, Clinical University Hospital of Santiago de Compostela, Health Research Institute of Santiago de Compostela, University of Santiago de Compostela, CIBERER, MetabERN, 15706 Santiago de Compostela, Spain; 3Unidad de Enfermedades Mitocondriales-Metabólicas Hereditarias, Servicio de Pediatría, Centro de Referencia Nacional (CSUR) y Europeo (MetabERN) para Enfermedades Metabólicas Hereditarias, Hospital Universitario 12 de Octubre, 28041 Madrid, Spain; 4Pediatric Gastroenterology, Hepatology and Nutrition Unit, Complejo Hospitalario Universitario Insular Materno-Infantil (CHUIMI), Universidad de Las Palmas de Gran Canaria, Asociación Canaria para La Investigación Pediátrica, Centro de Investigación Biomédica en Red de la Fisiopatología de la Obesidad y la Nutrición (CIBEROBN) ISCIII, 35016 Gran Canaria, Spain; 5Laboratory of Metabolic Disorders and Newborn Screening, Institute of Biomedical Research in Málaga (IBIMA-Plafatorma BIONAND), IBIMA-RARE, Málaga Regional University Hospital, 29010 Málaga, Spain; 6Nutrition and Metabolophaties Unit, Hospital Universitario La Fe, 46026 Valencia, Spain

**Keywords:** allelic expression imbalance, differential gene expression, inherited metabolic disorders, multi-omics, targeted transcriptomics

## Abstract

Inborn errors of metabolism (IEM) constitute a huge group of rare diseases affecting 1 in every 1000 newborns. Next-generation sequencing has transformed the diagnosis of IEM, leading to its proposed use as a second-tier technology for confirming cases detected by clinical/biochemical studies or newborn screening. The diagnosis rate is, however, still not 100%. This paper reports the use of a personalized multi-omics (metabolomic, genomic and transcriptomic) pipeline plus functional genomics to aid in the genetic diagnosis of six unsolved cases, with a clinical and/or biochemical diagnosis of galactosemia, mucopolysaccharidosis type I (MPS I), maple syrup urine disease (MSUD), hyperphenylalaninemia (HPA), citrullinemia, or urea cycle deficiency. Eight novel variants in six genes were identified: six (four of them deep intronic) located in *GALE*, *IDUA*, *PTS*, *ASS1* and *OTC*, all affecting the splicing process, and two located in the promoters of *IDUA* and *PTS*, thus affecting these genes’ expression. All the new variants were subjected to functional analysis to verify their pathogenic effects. This work underscores how the combination of different omics technologies and functional analysis can solve elusive cases in clinical practice.

## 1. Introduction

The diagnosis of Mendelian diseases has been transformed by next-generation sequencing (NGS). In the past, determining the genetic cause of a patient’s condition was a laborious task; genome sequencing has made it much easier [1].

Exome sequencing, whole exome sequencing (WES), and clinical exome sequencing (CES) are first choice technologies for the genetic diagnosis of patients with inborn errors of metabolism (IEM) detected either by newborn screening (NBS) or after the appearance of clinical symptoms. A lack of diagnosis can, however, happen, usually due to limitations in variant prioritization [2] and the inability to detect variants in non-coding areas of the genome, including those found in regulatory regions, deep intronic mutations, epivariations, and structural variations [3,4,5,6,7]. Whole genome sequencing (WGS) is an alternative that provides complete coverage of the genome and which is more sensitive than WES with respect to variants involving indels, chromosomal rearrangements and copy number variations [8]. WGS can also detect trinucleotide expansions (a recently reported cause associated with IEM) [9]. Nevertheless, the prioritization of detected variants remains challenging given the vast number of changes detected and our very incomplete knowledge of the effect of variants in non-coding sequences. Indeed, with WGS, the diagnostic rate only increases to 40% [10].

Exome sequencing takes into account less than 2% of the genome. Some of the remainder is involved in the regulation of expression in the exome, and many genetic defects associated with the dysregulation of expression are known. The main variants responsible for these alterations are those that interfere with proximal promoters, variants in the 5′ and 3′ untranslated regions (UTR), epivariations (DNA methylation defects), enhancer variants and structural variations [5,6,7,11,12,13,14,15,16]. One proposal aimed at increasing the genetic diagnostic rate contemplates combining genomic analysis and RNA sequencing (RNA-Seq) [17]. This would allow the effect of most variant types to be analyzed. Indeed, RNA-Seq detects altered gene expression, allelic expression imbalances (AEI), and aberrant transcripts. Once such a defect is identified, genomic sequencing can be targeted towards the genes selected by mRNA analysis with the aim of identifying the genetic root of the problem [17,18]. In IEM, the identification of a biomarker would also allow for the mRNA of a specific gene or a specific panel of genes (in heterogeneous disorders) to be examined. The challenge of reaching a confirmatory genetic diagnosis for patients affected by an IEM might therefore be met by a multi-omics pipeline that combines transcriptomic, genomic and metabolomic analyses. This paper shows how the use of a targeted diagnostic pipeline combining multiple omics technologies with functional analysis can reduce the diagnostic gap in IEM.

## 2. Results

### 2.1. Cases Report

The study included six unsolved patients. The six patients (P1, P2, P3, P4, P5 and P6) were clinically and/or biochemically diagnosed with galactosemia, mucopolysaccharidosis type I (MPS I), maple syrup urine disease (MSUD), hyperphenylalaninemia (HPA), citrullinemia and ornithine transcarbamylase (OTC) deficiency, respectively (Table 1). For P1, P2 and P5, enzymatic activity studies were also performed. A 96% decrease in UDP-Galactose-4-Epimerase (GALE) activity, a 100% decrease in α-L-Iduronidase (IDUA) activity and a 70% decrease in Argininosuccinate Synthase 1 (ASS1) activity were detected in P1, P2 and P5, respectively.

### 2.2. Exomic Studies

After clinical/biochemical diagnosis, exomic studies were performed on blood-extracted DNA via NGS, using either CES or WES. In P1, a previously described [19] likely pathogenic variant (c.284G>A) in the maternal allele of *GALE* was identified. In P2, a novel splicing variant (c.1524+2T>A) was detected in *IDUA*, predicted to be pathogenic according to the American College of Medical Genetics and Genomics (ACMG) criteria. Segregation studies confirmed that the variant was in the paternal allele. In P3, a previously described [20] likely pathogenic variant (c.827T>G) was detected in the maternal allele of *DBT*. No candidate variants were detected in P4 and P5. Mean coverage analysis suggested there were no deletions in any gene. After these studies, all patients remained with an inconclusive genetic diagnosis (Table 2).

### 2.3. Transcriptomic Studies

RNA-Seq was used to complete the diagnosis of P1, P2, P3, P4 and P5. An insufficient amount of RNA prevented from performing RNA-Seq on P6. The analysis was targeted to specific genes based on biochemical or enzymatic findings. For patients with a previously identified pathogenic variant, the expression of one allele was compared against the other. P1, P2 and P3 showed an AEI of the variants identified in *GALE*, *IDUA* or *DBT*, respectively (Figure 1a). For P4 and P5, in which no candidate variants had been identified, the expression analyses of the genes associated with HPA and citrullinemia were analyzed, respectively. No expression changes were observed for the genes associated with HPA in P4 compared to healthy controls (Figure 1b). For P5, a reduction in *ASS1* mRNA expression was identified (Figure 1b); this was confirmed by retro-transcriptase quantitative PCR (RT-qPCR), which revealed an 84% reduction in *ASS1* transcripts compared to healthy controls (Figure 1c).

Aberrant transcripts were also studied. RNA-Seq identified an insertion of 97 bp between exons 1 and 2 of *PTS* (one of the genes associated with HPA) in P4 in approximately 50% of reads (Figure 1d). The inserted region was a fragment of the intron found between exons 1 and 2. Since the pseudo-exon changes the reading frame of *PTS*, this mRNA should be degraded by nonsense-mediated decay (NMD) and, therefore, should be under-detected compared to the normal allele. As the number of reads of both alleles was similar, normal PTS expression was assessed by RT-qPCR; a 66% reduction in normal transcripts was seen in P4 compared to healthy controls (Figure 1c). An expression defect in this allele is therefore likely.

### 2.4. Genomic Studies

In order to identify the genetic causes of the expression or splicing defects detected by RNA-Seq, a hierarchical pipeline strategy was followed in which the proximal promoter and the 3’ UTR were first sequenced. The variants found were prioritized according to their minor allele frequency (MAF), their zygosity, and disease segregation. For P1, the variant c.-77G>C in the last nucleotide of exon 1 (non-coding) in *GALE* was prioritized; for P2, the variant c.-87T>C in *IDUA* was selected; for P4, the variant c.-82_-71delins-103_-86 detected in *PTS* was chosen. All these variants were present in heterozygosis. The variants found in P1 and P2 were in trans (Appendix A) with the pathogenic changes previously detected in DNA studies. All these variants are novel and classified as variants of uncertain significance (VUS).

In P4, in which a pseudo-exon insertion was identified, the intronic sequences flanking the inserted region were amplified and sequenced. This allowed the identification and prioritization of the variants c.83+658C>G and c.83+758T>A in *PTS* in P4 (both in the maternal allele and in trans with c.-82_-71delins-103_-86) (Appendix A). In P3 and P5, in which an AEI and reduced expression were detected, respectively, by RNA-Seq, WGS allowed the identification and prioritization of a deep intronic change in both patients. In P3, we identified the previously described [21,22] pathogenic variant c.1018-550A>G in *DBT* in heterozygosis and in trans with the variant identified by CES (Appendix A). In P5, the genomic sequencing identified and prioritized the novel change c.598-757G>A in homozygosis in *ASS1*. Both variants presumably cause a pseudo-exon insertion. Re-evaluation of RNA-Seq data showed an insertion between exons 8 and 9 of *DBT* in P3 (never detected before by specific cDNA analysis), and between the same exons of *ASS1* in P5 (Figure 1d). Both pseudo-exons were mapped in a small number of reads (two for the pseudo-exon found in P3 and seven for the one identified in P5), explaining why they escaped earlier analysis.

In P6, a novel, de novo, deep intronic variant (c.541-277A>G) was identified in heterozygosis in *OTC*, classified by ACMG as a VUS. In silico studies suggested an increase in the strength of an existing donor splice site (5′ GT; analyzed by Alamut Visual Plus Software v1.4).

### 2.5. Specific cDNA Analysis

After bioinformatics analysis, the effect of all novel variants suggestive of causing a splicing defect on cellular cDNA was next analyzed. RT-PCR was performed on fibroblast-extracted RNA from P1, P4 and P5, and from liver biopsy-extracted RNA from P6. Specific primers for *GALE*, *PTS*, *ASS1* and *OTC* were used, respectively. The pathogenic role of all four variants was then confirmed. The c.-77G>C in *GALE* led to an insertion of 110 nucleotides between exons 1 and 2 in P1 (not present in controls) (Figure 2a). This insertion was not detected via the RNA-Seq data, probably due to the reduced presence of this transcript in P1, and the existence of several transcripts of *GALE* without this exon. In P4, the pseudo-exon insertion detected by RNA-Seq was confirmed by specific cDNA analysis, which revealed a pseudo-exon insertion of 97 bp between exons 1 and 2 of *PTS* that was not present in healthy controls (Figure 2b). In the re-evaluation of the RNA-Seq data using Integrative Genomics Viewer (IGV) software v2.8.13, a pseudo-exon insertion was observed between exons 8 and 9 of *ASS1* in P5 (Figure 2c). It was also present in the control population but at a much lower level since variant c.598-757G>A increases the strength of the recognition of a pre-existing donor splice site. Finally, in P6, the amplification of a cDNA fragment containing exons 5 and 6 revealed a 57 bp pseudo-exon insertion flanked by the variant at its 3′ end (Figure 2d). Although this variant was not inherited from the parents, the study of a single nucleotide polymorphism present in the pseudo-exon identified its linkage to the paternal allele (Appendix A). The HUMARA test could not be performed on liver biopsy samples, but in the blood samples, it revealed a random inactivation of the X chromosome.

Together, these results suggest that the variants c.-77G>C, c.[83+658C>G;83+758T>A], c.598-757G>A and c.541-277A>G found in *GALE*, *PTS*, *ASS1* or *OTC*, respectively, are responsible for the splicing defects found in these genes in P1, P4, P5 and P6, respectively.

### 2.6. Minigene Analysis

To isolate the new splicing variants from their genomic context, and therefore confirm them to be responsible for the cDNA defects seen, c.-77G>C (*GALE*), c.83+658C>G and c.83+758T>A (*PTS*), c.598-757G>A (*ASS1*) and c.541-277A>G (*OTC*) minigene studies were performed. The results suggest that the variant found in *GALE* leads to aberrant splicing that produces a fragment without the cloned region (Figure 3a). The combination of both variants found in *PTS*, as well as the variants found in *ASS1* and *OTC*, leads to an aberrant inclusion of the pseudo-exons previously found in the cDNA of P4, P5 and P6 (Figure 3b–d). These results thus suggest that the variants c.-77G>C, c.[83+658C>G; 83+758T>A], c.598-757G>A and c.541-277A>G found in *GALE*, *PTS*, *ASS1* and *OTC*, respectively, affect the splicing process, strongly implying them to be pathogenic.

### 2.7. Gene Expression Analysis by Luciferase Reporter Assay

To study the potential effect of c.-87T>C and c.-82_-71delins-103_-86 on *IDUA* and *PTS* expression, and to rule out a potential effect of c.-77G>C on the promoter of *GALE*, a firefly luciferase-coding sequence was cloned under the promoter of the three genes. The amplified regions were selected by attending to different databases (see Materials and Methods). Two proximal promoter regions were identified for *GALE* and *PTS*, and one for *IDUA* (Figure 4a). In all cases, luciferase was expressed, but an 87% reduction was recorded in the context of the mutant *IDUA* promoter compared to the wild-type promoter, and a 59% reduction in the context of the mutant *PTS* promoter; no differences were seen for the contexts of the mutant and wild-type *GALE* promoters (Figure 4b). These results indicate that the variants found in the 5′ UTR of *IDUA* (c.-87T>C) and *PTS* (c.-82_-71delins-103_-86) in P2 and P4, respectively, are probably responsible for the expression defects seen. In contrast, the variant found in P1 in *GALE* (c.-77G>C) does not appear to affect the gene promoter.

## 3. Discussion

NGS has had a significant impact on the diagnosis of rare diseases, especially heterogeneous genetic diseases and those with unclear phenotypes [23]. Genetic confirmation, even in IEM, is essential for appropriate genetic counseling and in some cases for the application of gene- or mutation-specific therapies. Notwithstanding, the diagnosis rate remains much lower (25–50%) than initially expected [18,24,25,26,27], leaving many patients without genetic confirmation of their condition. This is partly the result of the incapacity of exome sequencing to identify non-coding variants [2]. The combination of other omics technologies and functional genomics might, however, help reduce the diagnostic gap in IEM, as previously described in other works [28].

In IEM, genetic diagnostic technologies can be used with more confidence since biomarkers are available for many disorders, helping to focus genetic analyses. The six patients in the present work had previously been diagnosed clinically and/or biochemically—but not genetically—as having galactosemia, MPS I, MSUD, HPA, citrullinemia or OTC deficiency. By combining the DNA and RNA-Seq analyses of specific genes, it became possible to identify variants outside of the exomic regions. Indeed, RNA-Seq allowed the identification of an allelic expression imbalance of *GALE*, *IDUA* and *DBT* in P1, P2 and P3, respectively. In addition, in P2, transcriptomic analysis confirmed the effect of the novel variant identified in exome sequencing (c.1524+2T>A): an insertion of four nucleotides between exons 10 and 11 in *IDUA* due to the disruption of the cryptic donor splicing site of exon 10.

The presence of aberrant transcripts in *PTS* in P4 helped to direct subsequent analyses. Although no expression defect was detected by RNA-Seq, the presence of the pseudo-exon insertion in the same number of reads as the normal allele was suggestive of one. Indeed, a novel 5′ UTR variant was identified by targeted DNA studies, and its effect on gene expression was confirmed by specific functional analysis.

Unfortunately, RNA-Seq detection suffers from the degradation of out-of-frame transcripts by NMD. This was apparent in P3 and P5, in which the novel isoforms were almost undetectable by RNA-Seq. However, gene expression analysis revealed a transcription defect in both cases. For these patients, WGS had to be used to identify the deep intronic changes responsible for their diseases. The use of NMD inhibitors might improve the detection of pseudo-exon inclusions. Taken together, these results suggest that, while a gene’s expression may not seem altered, nor may any aberrant transcripts be detectable by RNA-Seq, the data collected should be re-analyzed since some alterations can escape such restrictive assessments.

The genomic sequencing of the 5′ non-coding regions allowed the identification and prioritization of c.-77G>C in *GALE*, c.-87T>C in *IDUA* and c.-82_-71delins-103_-86 in *PTS*, as candidate variants behind the expression defects in P1, P2 and P4, respectively. Note that variant c.-77G>C was not detected in exome sequencing despite its localization in the first exon of *GALE*. Since this exon is non-coding, it is not covered by the targets designed for CES. The present results underscore the importance of sequencing all exons (coding or non-coding) in targeted exome sequencing.

When a novel variant is identified, functional studies are necessary to determine if it has a pathogenic effect. The tests needed must be designed ad hoc depending on the proposed effect of the variant. The present work identified variants affecting either expression or splicing. To study the former, a firefly luciferase reporter assay was used, and for the latter, minigene functional studies. These analyses confirmed the pathogenic effect of two variants affecting the expression of *IDUA* and *PTS*, and of five variants affecting the splicing process of *GALE*, *PTS*, *ASS1* and *OTC*. The c.-77G>C variant found in *GALE*, predicted to affect splicing, was also found in the promoter region of the gene. The gene expression was not altered by this variant, but its presence in the non-coding exon 1 of *GALE* affected its splicing, leading to the retention of a part of intron 1. Since this insertion occurs before the AUG translation initiation codon, it should not change the reading frame. It may be that a miss-splicing of intron 1 causes a failure in Transcription-Export (TREX) binding. TREX is the complex responsible for mRNA export from the nucleus to the cytoplasm [29]. Thus, intron retention by *GALE* would keep the aberrant transcript in the nucleus, precluding its translation and reducing the amount of coded protein [30].

In conclusion, the present work identifies and confirms the pathogenic role of eight new variants in different genes. It also shows the importance of using techniques complementary to WES and demonstrates the diagnostic strength of multi-omics technologies used in combination with functional studies. Finally, in agreement with other studies that endorse the use of WGS over WES, the results highlight the importance of analyzing non-coding regions.

## 4. Materials and Methods

### 4.1. DNA Studies

High-purity DNA was extracted from peripheral blood or from patient-derived fibroblasts using the MagNA Pure Compact System and either the MagNA Pure Compact Nucleic Acid Isolation Kit I-Large Volume or the MagNA Pure Compact Nucleic Acid Isolation Kit I (Roche Applied Science, Indianapolis, IN, USA) following the manufacturer’s instructions.

The variants responsible for the patients’ clinical/biochemical findings were identified using: (1) a targeted metabolic panel covering the exome of 119 genes (Nextera Nature Capture [Illumina, San Diego, CA, USA]) and the entire sequence of *PAH*, *ALDOB*, *OTC*, *SLC22A5*, *GLDC* and *PCCA* [31], (2) a targeted panel including 4813 genes (Clinical-Exome Sequencing TruSight™ One Gene Panel [Illumina, San Diego, CA, USA]), (3) WES (TruSeq Rapid Exome Library Prep Kit [Illumina, San Diego, CA, USA]) or (4) WGS (KAPA HyperPrep Kit [Roche Applied Science, Indianapolis, IN, USA]), following the manufacturer’s instructions. Libraries were sequenced using a NextSeq 500 Mid Output Kit (Illumina, San Diego, CA, USA) for the panels and WES libraries, or a NovaSeq 6000 (Illumina, San Diego, CA, USA) for the WGS libraries.

The variants identified were analyzed using different bioinformatic tools. Those found in coding regions were prioritized by the TruSight Software Suite v2.5 (Illumina, San Diego, CA, USA) using different filters. Those variants with a MAF of >0.5% in the Genome Aggregation Database (gnomAD; http://gnomad.broadinstitute.org/ accessed on 22 April 2022) [32] were excluded. The remaining variants were prioritized according to: (1) the correlation between the candidate gene and the patient’s clinical and/or biochemical data, (2) whether they produced a loss of function in the protein, (3) whether they are described in the Human Gene Mutation Database (HGMD Professional release 2022.2, https://portal.biobase-international.com/hgmd/pro/start.php, accessed 22 April 2022), ClinVar [33] and/or the Leiden Open Variation Database (LOVD) [34] as pathogenic or likely pathogenic according to ACMG guidelines [35] and (4) if they were predicted to be pathogenic or likely pathogenic by the VarSome platform (which comprises 17 different predictors) (https://varsome.com/, accessed 22 April 2022) [36] or by Alamut Visual Plus Software (v.1.4) (which comprises the Align GVGD, SIFT and MutationTaster programs) (SOPHiA GENETICS, Lausanne, Switzerland).

Variants in exon-intron boundaries returning an MAF of >0.5% were excluded. The candidates were then analyzed in silico to determine their potential effect on splicing using Alamut Visual Plus (v.1.4) (SOPHiA GENETICS, Lausanne, Switzerland) and Human Splicing Finder v3.1 (Genomnis, Marseille, France) software [37].

Segregation studies were performed by amplification of parental blood-extracted DNA by PCR using FastStart Taq DNA Polymerase (Roche Applied Science, Indianapolis, IN, USA) and Sanger sequencing using the BigDye Terminator Cycle Sequencing Kit (Applied Biosystems, Foster City, CA, USA).

### 4.2. RNA-Seq

RNA-Seq was performed for P1, P2, P3, P4, P5 and 15 healthy controls. Patient fibroblasts were obtained from skin biopsies, and the control cells from different fibroblast cell lines were obtained from either the Coriell Institute for Medical Research (Camden, NJ, USA) or Lonza Biotech (Basel, Switzerland). Cultures were maintained in Minimal Essential Medium (MEM) supplemented with 10% fetal calf serum, 1% glutamine, 100,000 U/L penicillin, and 100 mg/L streptomycin. Cells were maintained in a humidified incubator held at 5% CO_2_. RNA was extracted using the RNeasy Micro Kit (Qiagen, Hilden, Germany) according to the manufacturer’s instructions. The concentration and quality of the obtained RNA were measured using a 2100 Bioanalyzer (Agilent Technologies, Santa Clara, CA, USA). Libraries were prepared using the TruSeq Stranded mRNA Library Prep Kit (Illumina, San Diego, CA, USA) and sequenced in a NovaSeq 6000 system (Illumina, San Diego, CA, USA), retrieving 100 bp paired-end reads. The data quality was assessed with FASTQC software (Galaxy Version 0.73), and the files were trimmed with FASTP. Mapping was performed with HISAT2, quantification files were obtained with featureCounts, and differentially expressed genes were examined using Limma software (Galaxy Version 3.50.1). RNA-Seq reads were visualized using IGV software v2.8.13.

### 4.3. Validation of Transcriptomic Data

Differential gene expression results were validated using RT-qPCR. These assays were performed starting with 250 ng of total RNA; single-stranded cDNA was obtained by retrotranscription using the NZY First-Strand cDNA Synthesis Kit (NZYTech, Lisbon, Portugal) following the manufacturer’s protocol. Specific primers were designed for each gene. *GUSB* was used as an endogenous control. qPCR experiments were performed in a LightCycler^®^ 480 Instrument (Roche Applied Science, Indianapolis, IN, USA) using PerfeCTa SYBR^®^ Green FastMix (Quantabio, Beverly, MA, USA), following the LightCycler^®^ manufacturer’s instructions except for the amplification step which was modified to 10 s at 95 °C, 30 s at 60 °C and 30 s at 72 °C. CT (cycle threshold) values were obtained and analyzed using the 2^−ΔΔCt^ method.

Aberrant splicing isoforms were confirmed by the conversion of 1500 ng of total RNA to cDNA using the SuperScript VILO cDNA Synthesis Kit (Thermo Fisher Scientific, Waltham, MS, USA) following the manufacturer’s protocol. Fragments of interest were then amplified with the corresponding primers by PCR using FastStart Taq DNA Polymerase (Roche Applied Science, Indianapolis, IN, USA) and Sanger sequenced using the BigDye Terminator Cycle Sequencing Kit (Applied Biosystems, Foster City, CA, USA).

### 4.4. Minigene Studies

To examine the splicing pattern in vitro, the pSPL3 vector was used (Exon Trapping System, Gibco, BRL, Carlsbad, CA, USA). Gene fragments corresponding to the intronic sequence containing the pseudo-exon located in intron 1 of *PTS*, intron 8 of *ASS1* and intron 5 of *OTC* in P4, P5 and P6 were amplified from patient DNA (Figure 3b–d) and cloned into the pGEMT-Easy vector (Promega, Madison, WA, USA); the alleles were isolated. The insertion was excised with *Eco*RI (Roche Applied Science, Indianapolis, IN, USA), purified using the QIAquick Gel Extraction Kit (Qiagen, Hilden, Germany) and subsequently cloned into the pSPL3 vector dephosphorylated with Thermosensitive Alkaline Phosphatase (Promega, Madison, WA, USA). Ligation was performed using the Rapid DNA Ligation Kit (Thermo Fisher Scientific, Waltham, MS, USA). Clones containing the desired normal and mutant insertions were identified by restriction enzyme analysis and automated DNA sequencing. A total of 2 µg of the wild-type or mutant minigene was then transfected into the human hepatic cell line Hep3B (supplied by Dr. S.R. de Córdoba) using JetPEI transfection reagent (Polyplus-Transfection, Illkirch, France) following the manufacturer’s protocol. Cells were harvested 48 h post-transfection.

To assess the functional effect of the c.-77G>C change identified in the last nucleotide of the exon 1 of *GALE*, a hybrid minigene was constructed for transfection experiments. Since the variant is located in the first exon of *GALE*, and, therefore, no 3′ splice-site is present, the minigene was constructed to contain the 3′ splice-site of another gene, namely exon 2 of *SPR* [38]. The entire sequence of *GALE* exon 1 was amplified using specific primers, adding the restriction targets for *Pfo*I and *Nhe*I. The obtained fragment was cloned into the pGEMT-Easy vector (Promega, Madison, WA, USA), excised with *Pfo*I (New England Biolabs, Beverly, MA, USA) and *Nhe*I (Roche Applied Science, Indianapolis, IN, USA), purified using the QIAquick Gel Extraction Kit (Qiagen, Hilden, Germany) and then cloned into the pSPL3-SPR vector (previously generated at our laboratory) using the Rapid DNA Ligation Kit (Thermo Fisher Scientific, Waltham, MS, USA) (Figure 3a). The clone selection and transfection were performed as described for the *PTS*, *ASS1* and *OTC* minigenes.

### 4.5. Luciferase Studies

The *GALE*, *IDUA* and *PTS* promoters and transcription start sites (TSS) were identified using the Eukaryotic Promoter Database (EPD) (https://epd.epfl.ch//index.php, accessed 12 January 2022) and the ENCODE Candidate Cis-Regulatory Elements (cCREs) Registry on the University of California Santa Cruz (UCSC) genome browser (https://genome.ucsc.edu/, accessed 13 January 2022).

The selected regions were amplified using specific primers, adding the restriction targets for *Kpn*I and *Bgl*II. Fragments were then cloned into the pGEMT-Easy vector (Promega, Madison, WA, USA) to separate the alleles. The insertion was excised with *Kpn*I (New England Biolabs, Beverly, MA, USA) and *Bgl*II (Roche Applied Science, Indianapolis, IN, USA), purified using the QIAquick Gel Extraction Kit (Qiagen, Hilden, Germany) and then cloned into the pGL3-Basic vector (Promega, Madison, WA, USA) using the Rapid DNA Ligation Kit (Thermo Fisher Scientific, Waltham, MS, USA). Clones containing the desired normal and mutant insertions were identified by restriction enzyme analysis and automated DNA sequencing.

The human hepatic cell line Hep3B (supplied by Dr. S.R. de Córdoba) was then co-transfected with 1.8 µg of the wild-type or mutant construct in addition to 0.2 µg pRL vector (supplied by Dr. I. Ventoso) using JetPEI transfection reagent (Polyplus-Transfection, Illkirch, France) following the manufacturer’s indications. Cells were harvested 48 h post-transfection.

Firefly and *Renilla reniformis* luciferase activities were assessed using the Dual-Luciferase Reporter Assay System (Promega, Madison, WA, USA) following the manufacturer’s indications, and detected using a FLUOstar OPTIMA microplate reader (BMG Labtech, Durham, NC, USA).

### 4.6. X-Chromosome Inactivation Studies (HUMARA Assay)

A fragment of the *AR* gene containing a highly polymorphic short tandem repeat was amplified using specific fluorescent primers and FastStart Taq DNA polymerase (Roche Applied Science, Indianapolis, IN, USA). The amplicon size of the patient, paternal and maternal samples was analyzed by capillary electrophoresis using a 3730xl DNA Analyzer (Applied Biosystems, Foster City, CA, USA). The results were processed using Peak Scanner Software 2 v.2.0 (Applied Biosystems, Foster City, CA, USA). Patient blood DNA was digested using either *Hpa*II (Thermo Fisher Scientific, Waltham, MS, USA), *Msp*I (Thermo Fisher Scientific) or a mock enzyme, PCR amplification was repeated and the amplicon size was determined in the same manner as above. The area under the peaks of *Hpa*II and the mock-treated samples were compared to obtain the methylation percentage of each allele.

### 4.7. Statistical Analysis

Statistical analyses were performed using GraphPad Prism 6 software (GraphPad Software, La Jolla, CA, USA) for Windows. Student’s *t*-test was used for comparison between groups. The significance was set at *p* < 0.05.

### 4.8. Accession Numbers

*GALE* (NM_001008216.2); *IDUA* (NM_000203.5); *DBT* (NM_001918.5); *PTS* (NM_000317.3); *ASS1* (NM_054012.4); *OTC* (NM_000531.6).

## Figures and Tables

**Figure 1 ijms-23-12850-f001:**
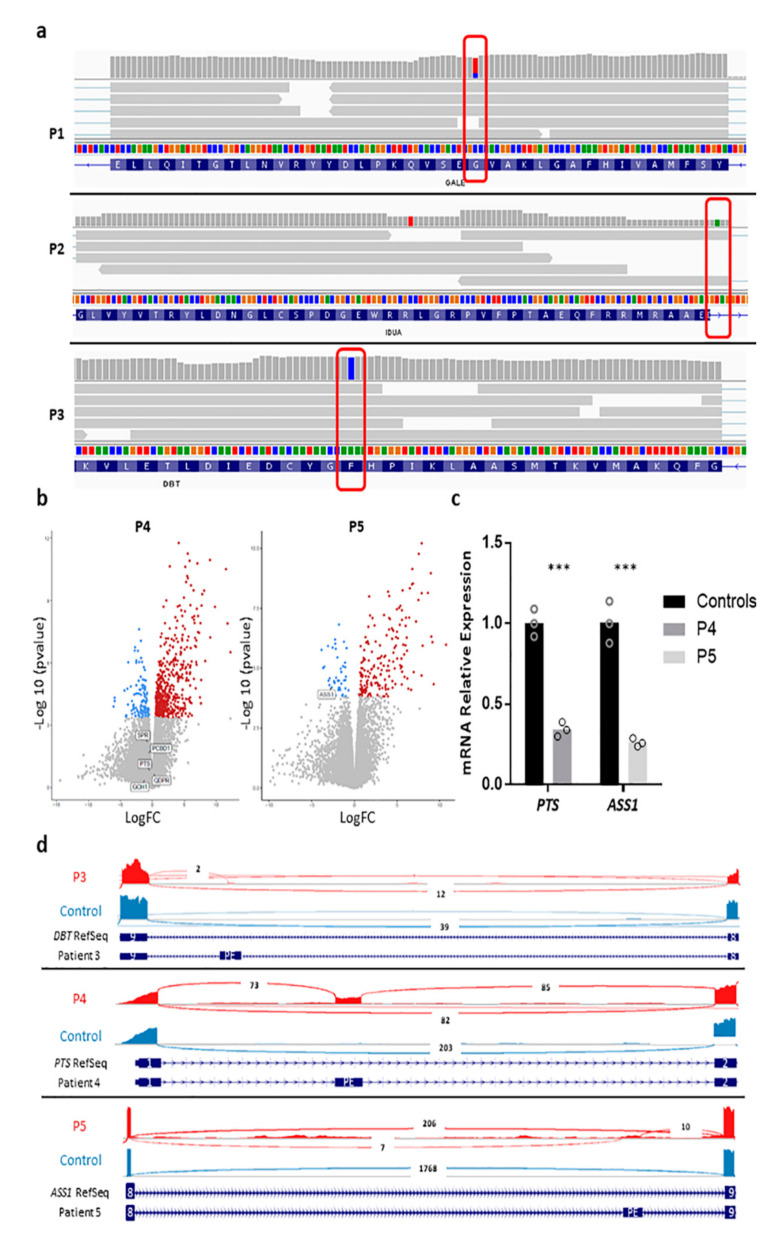
RNA-Seq analysis. (**a**) RNA-Seq reads mapped onto the *GALE* gene in P1 (top panel); variant c.284G>A is marked in a red rectangle. The wild-type sequence is shown in blue (14%) and the mutant in red (86%). The middle panel shows the mapped RNA-Seq reads for the *IDUA* gene in P2. The pathogenic variant c.1524+2T>A is marked in a red rectangle; the insertion of 4 bp at the end of exon 10 is shown. The lower panel shows mapped RNA-Seq reads for *DBT* in P3. The pathogenic variant c.827T>G is shown in a red rectangle. Visualization of reads obtained via the Integrative Genomics Viewer v2.8.13 (IGV). (**b**) Volcano plots for P4 and P5. Genes associated with hyperphenylalaninemia are marked for P4; *ASS1* is labeled for P5. Significantly upregulated genes are shown in red, downregulated in blue, and those showing no significant change in grey. Plots were obtained using https://usegalaxy.eu, accessed on 21 April 2022. (**c**) RT-qPCR for P4 and P5 for the *PTS* or *ASS1* genes, respectively. Results are from three different experiments using samples from two healthy controls. Each circle represents the mean of three different replicates (*** *p* < 0.001). (**d**) Sashimi plot of the pseudo-exon (PE) insertion detected by RNA-Seq in *DBT* in P3 (upper panel), the PE observed in *PTS* in P4 (middle panel), and the PE detected in *ASS1* in P5 (lower panel). The gene model for the RefSeq annotation is shown in blue; the inserted PE is shown at the bottom. The number of RNA-Seq split reads is indicated in the exon-connecting line. Sashimi plots were created using IGV and the University of California Santa Cruz (UCSC) genome browser.

**Figure 2 ijms-23-12850-f002:**
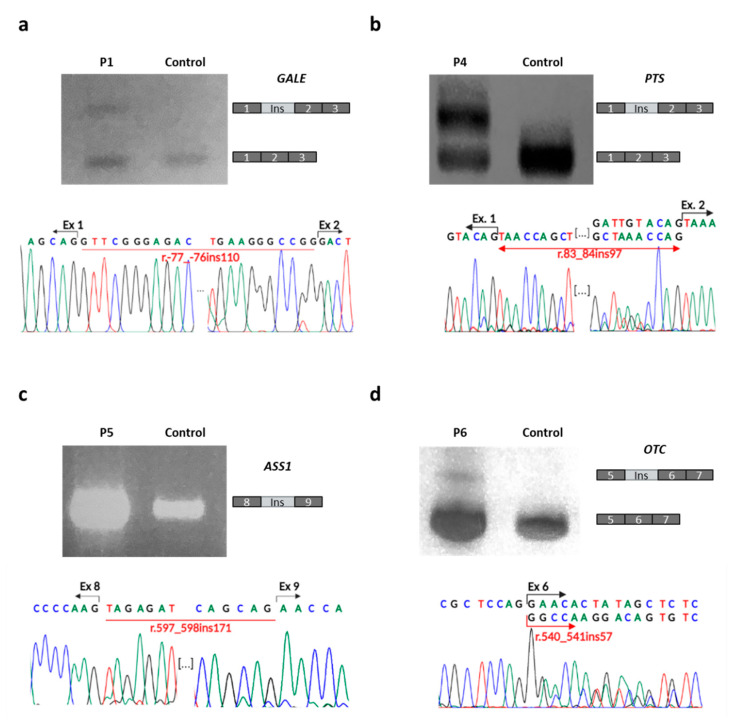
Specific cDNA analyses. Amplification of a small fragment of cDNA of *GALE* in P1 (**a**), of *PTS* in P4 (**b**), of *ASS1* in P5 (**c**) and of *OTC* in P6 (**d**). The figure shows agarose gels for each amplification, comparing each patient with a healthy control, plus the sequence of each amplicon.

**Figure 3 ijms-23-12850-f003:**
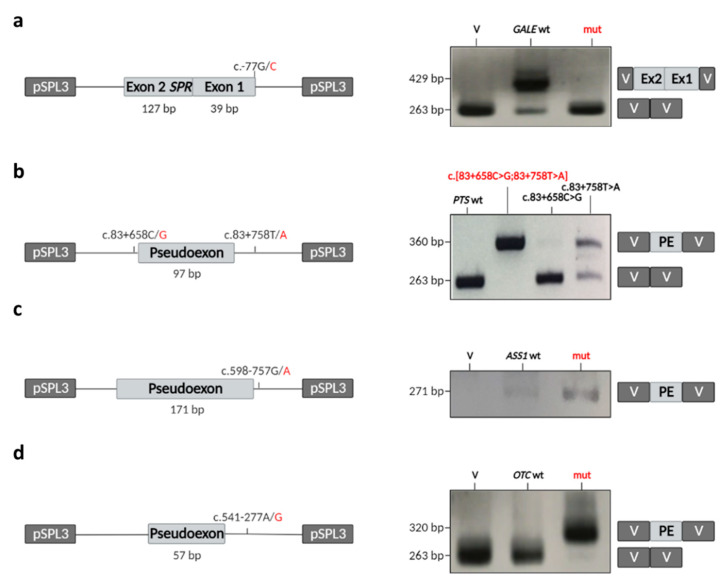
Transcriptional analysis of minigene-derived (*GALE*, *PTS*, *ASS1* and *OTC*) splicing variants. RT-PCR analysis of (**a**) *GALE* wt or c.-77G>C, (**b**) *PTS* wt, c.[83+658C>G;83+758T>A] or each variant on its own, (**c**) *ASS1* wt or c.598-757G>A and (**d**) *OTC* wt or c.541-277A>G. A diagram of the cloned region is shown in the left panel with variant alleles marked in red. The right side of the figure shows the agarose gel electrophoresis results for the amplified fragments, plus the inserted regions. All amplifications were performed with internal vector primers, except for *ASS1*, which was amplified with a reverse primer spanning the PE-V junction. V: vector, Ex: exon, PE: pseudo-exon.

**Figure 4 ijms-23-12850-f004:**
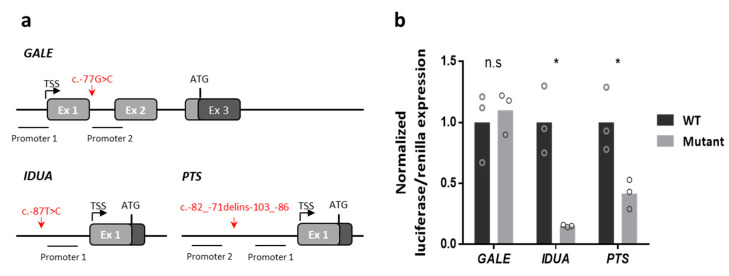
Functional study of 5′ UTR variants. (**a**) The proximal promoters identified for the three genes are shown, plus the candidate variants for altering gene expression, the transcription start site (TSS), and the ATG translation initiation codon. Coding exons and non-coding exons are represented in dark and light grey, respectively. Promoter numbering follows that of the Eukaryotic Promoter Database (EPD). (**b**) Normalized luciferase/*renilla* expression of the wild-type or mutant promoter of *GALE*, *IDUA* and *PTS*. Results are from three different experiments. Each circle represents the mean of three different replicates (* *p* < 0.05). n.s: non-significant.

**Table 1 ijms-23-12850-t001:** Patient cohort. Biochemical data at diagnosis, clinical/biochemical diagnosis and age at initial diagnosis.

Patient	Sex	Biochemical/Clinical Data at Diagnosis	Clinical/Biochemical Diagnosis	Age at Initial Diagnosis
P1	Female	Galactitol (urine): 462 mmol/mol creat. Galactonate (urine): 328 mmol/mol creat. Galactose-1-P (erythrocytes): 4.8 µmol/g hemoglobin Elevated hepatic transaminase (HP:0002910) Bilateral congenital cataracts (HP:0000519) Dilated cardiomyopathy (HP:0001644) GALT activity: normal GALK activity: normal GALE activity: 4%	Galactosemia	4 months
P2	Male	Chondroitin sulfate (urine): 5.4 mg/mmol creat. Dermatan sulfate (urine): 5.5 mg/mmol creat. Heparan sulfate (urine): 37 mg/mmol creat. Keratan sulfate (urine): 7.4 mg/mmol creat. Constrictive median neuropathy (HP:0012185) Limitation of joint mobility (HP:0001376) Joint stiffness (HP:0001387) IDUA activity: undetectable	Mucopolysaccharidosis	5 years
P3	Male	Valine (plasma): 581.2 µmol/L Leucine (plasma): 649.1 µmol/L Isoleucine (plasma): 233.8 µmol/L Allo-isoleucine (plasma): 128.4 µmol/L Gait disturbance (HP:0001288) Unsteady gait (HP:0002317) Frequent falls (HP:0002359) Leukodystrophy (HP:0002415) Generalized hypotonia (HP:0001290) Gait ataxia (HP:0002066)	Maple syrup urine disease (MSUD)	1 year
P4	Female	Phenylalanine (blood): 5.6 mg/dL Neopterin (urine): 12.9 mmol/mol creat. Biopterin (urine): 0.3 mmol/mol creat. Biopterin (cerebral spinal fluid): normal Dihidropterine reductase activity: normal	Hyperphenylalaninemia (HPA)	Newborn screening program
P5	Male	Citrulline (urine): 61 mmol/mol creat. ASS1 activity: 30%	Citrullinemia	Newborn screening program
P6	Female	Orotic acid (urine): 418 mmol/mol creat. Glutamine: 1051 µmol/L Ammonia: 121 µmol/L Acute hepatic failure (HP:0006554) Elevated hepatic transaminase (HP:0002910) Abnormality of coagulation (HP:0001928)	Ornithine transcarbamylase (OTC) deficiency	15 months

Creat: creatinine. Normal values: galactitol: 40 ± 24 mmol/mol creat.; galactonate: 90 ± 43 mmol/mol creat.; galactose-1-P: <0.17 µmol/g haemoglobin; chondroitin sulfate: 6.1 ± 3.4 mg/mmol creat.; dermatan sulfate: 2.4 ± 0.7 mg/mmol creat.; heparan sulfate: 9.4 ± 5.3 mg/mmol creat.; keratan sulfate: 12.9 ± 7.6 mg/mmol creat.; valine: 184 ± 47 µmol/L; leucine: 99 ± 32 µmol/L; isoleucine: 50 ± 16 µmol/L; allo-isoleucine: indetectable; phenylalanine: <2 mg/dL; neopterin (urine): 1.2–12 mmol/mol creat.; biopterin (urine): 0.5–5.2 mmol/mol creat.; citrulline: 8 ± 10 mmol/mol creat.; orotic acid (urine): 3–9 mmol/mol creat.; glutamine: 427 ± 104 µmol/L; ammonia: <90 µmol/L.

**Table 2 ijms-23-12850-t002:** Genetic findings. Affected gene, variants identified by exomic or genomic sequencing, mRNA effect and age at definitive diagnosis.

Patient	Gen	Exomic Studies	Genomic Studies/RNA-Seq	mRNA Effect	Age at Definitive Diagnosis
P1	*GALE*	c.284G>A (maternal)	c.-77G>C (paternal)	Allelic expression imbalance	8 years
P2	*IDUA*	c.1524+2T>A (paternal)	c.-87T>C (maternal)	Allelic expression imbalance	8 years
P3	*DBT*	c.827T>G (maternal)	c.1018-550A>G (paternal)	Allelic expression imbalance	28 years
P4	*PTS*	-	c.83+658C>G; c.83+758T>A (maternal) c.-82_-71delins-103_-86 (paternal)	Aberrant transcripts Reduced expression	12 years
P5	*ASS1*	-	c.598-757G>A (homozygosis)	Aberrant transcripts	4 years
P6	*OTC*	-	c.541-277A>G (de novo)	Aberrant transcripts	5 years

## Data Availability

Not applicable.

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
