# Peer review of "Identification of Clinical Variants beyond the Exome in Inborn Errors of Metabolism"

_ijms, 2022, doi:10.3390/ijms232112850_

Round 1

Reviewer 1 Report

I regret that I am not sufficiently cognate to be able to provide feedback on the bioinformatic and laboratory analyses described. Consequently, my feedback (which I hope is helpful) is somewhat restricted. My main general question concerns what seems to me to be a key issue that has not been addressed adequately, namely how many of the 8 novel variants were discovered without resorting to the strategy that is the central theme of this MS?

49-52: given that the word “diagnosis’ in the present context means a diagnosis based on the identification of an actual causal variant by sequencing, it is not clear how the causes listed in this sentence could lead to misdiagnosis, which (in the same present context) must mean an actual diagnosis, but an incorrect diagnosis, which can only be based on identification of a variant other than the correct causal variant. To my understanding, the context described in this sentence would be far more likely to lead to a lack of diagnosis than to a misdiagnosis, i.e. to a failure to find any likely causal variant.

55-56: I am having trouble understanding the claim that trinucleotide expansions are “a recently reported cause of disease”. For example, the cause of Fragile-X syndrome was shown to be trinucleotide expansions back in the early 1990s.

58: the mention of “diagnostic rate” in this line reinforces my feedback in relation to lines 49-52, because a diagnostic rate of 40% does not mean a misdiagnostic rate of 60%. Surely a large proportion of the 60% will be failures of diagnosis.

65: is there a reference for this proposal?

77: I think it would be helpful if a more standard terminology could be used in relation to diagnosis. In line 58 we have “diagnostic rate”; in line 66 we have “diagnostic yield” and now we have “diagnostic gap”. It would be really helpful if the same term could be used in each of these contexts and elsewhere in the MS, e.g. the first paragraph of the Discussion.

84: a trivial issue: since the reference to Table 1 could just as validly have been inserted at the end of the previous sentence, and since the table includes so many different types of results, it would be best to move the first mention of the table to the end of the first sentence in this paragraph.

88: Table 1: given the stress on the novelty of the use of WGS in this MS, I am having trouble understanding why it is pooled with sequencing via the metabolic panel, which is a very different strategy. It would surely be far better if different symbols could be used for these two very different strategies. Also, it would be very helpful if the variants discovered in this study could be distinguished from the previously described variants in Table 1.

97-108: section 2.2 seems to suggest that 4 of the 8 novel variants reported in this MS were discovered via the technologies that were highlighted as being inadequate in the Introduction, without any resort to WGS. In fact, it seems from lines 265-267 that WGS was required only in P3 and P5.

103-104 & 162: can we please have a reference for each of the previously described variants? Also, I think it would be helpful to compare the strategy with which they were previously detected with the detection strategy in this study.

244: the wording here, namely “leaving many patients without genetic confirmation of their condition” reinforces my feedback on lines 49-52. The problem is lack of diagnosis, not misdiagnosis.

245-247: I am not sufficiently cognate to know of previous relevant literature, but I would be most surprised if no-one has ever suggested such a strategy before. In other words, are the authors certain that it is valid for them to imply that they are the first to suggest this strategy?

Author Response

I regret that I am not sufficiently cognate to be able to provide feedback on the bioinformatic and laboratory analyses described. Consequently, my feedback (which I hope is helpful) is somewhat restricted. My main general question concerns what seems to me to be a key issue that has not been addressed adequately, namely how many of the 8 novel variants were discovered without resorting to the strategy that is the central theme of this MS?

We thank Reviewer #1 for the careful revision of the manuscript.

We understand the Reviewer’s concern about the proper way of addressing the 8 novel variants and have tried to explain it more clearly:

Out of the 8 novel variants identified, 7 of them were discovered thanks to the multi-omics strategy (c.541-277A>G found in OTC which may be the most difficult to understand, it was found thanks to a metabolic panel that includes the intronic region of several genes, therefore not being considered as an exomic approach and falling into our suggestion).

49-52: given that the word “diagnosis’ in the present context means a diagnosis based on the identification of an actual causal variant by sequencing, it is not clear how the causes listed in this sentence could lead to misdiagnosis, which (in the same present context) must mean an actual diagnosis, but an incorrect diagnosis, which can only be based on identification of a variant other than the correct causal variant. To my understanding, the context described in this sentence would be far more likely to lead to a lack of diagnosis than to a misdiagnosis, i.e. to a failure to find any likely causal variant.

We appreciate the Reviewer’s indication and totally agree with it. “Misdiagnosis” refers to a different idea therefore we have changed it to make clear that we refer to a lack of diagnosis and not an incorrect one.

55-56: I am having trouble understanding the claim that trinucleotide expansions are “a recently reported cause of disease”. For example, the cause of Fragile-X syndrome was shown to be trinucleotide expansions back in the early 1990s.

The annotation pointed out by the Reviewer is quite certain. We realized that our sentence was incomplete. For this reason, we have changed it to “WGS can also detect trinucleotide expansions (a recently reported cause associated to IEM)” (lanes 60 and 61)

58: the mention of “diagnostic rate” in this line reinforces my feedback in relation to lines 49-52, because a diagnostic rate of 40% does not mean a misdiagnostic rate of 60%. Surely a large proportion of the 60% will be failures of diagnosis.

Following the feedback also stated in relation to lines 49-52 we have changed it across the manuscript to make it more understandable.

65: is there a reference for this proposal?

Yes, we have included it in the text. It is doi:10.1038/s41591-019-0457-8 (line 72).

77: I think it would be helpful if a more standard terminology could be used in relation to diagnosis. In line 58 we have “diagnostic rate”; in line 66 we have “diagnostic yield” and now we have “diagnostic gap”. It would be really helpful if the same term could be used in each of these contexts and elsewhere in the MS, e.g. the first paragraph of the Discussion.

Following the Reviewer’s advice, we have integrated “diagnostic rate” and “diagnostic yield” and always used the former since both refer to the same concept. In the case of “diagnostic gap” this is a different idea that corresponds to the differences across patients and diseases and does not mean a total number/percentage (that, would be the diagnostic rate).

84: a trivial issue: since the reference to Table 1 could just as validly have been inserted at the end of the previous sentence, and since the table includes so many different types of results, it would be best to move the first mention of the table to the end of the first sentence in this paragraph.

We thank the Reviewer for the correction. We have changed the location of the reference of Table 1. Also, please note that the table has been split in two tables for better understanding.

88: Table 1: given the stress on the novelty of the use of WGS in this MS, I am having trouble understanding why it is pooled with sequencing via the metabolic panel, which is a very different strategy. It would surely be far better if different symbols could be used for these two very different strategies. Also, it would be very helpful if the variants discovered in this study could be distinguished from the previously described variants in Table 1.

The pooling of WGS with the metabolic panel is due to the coverage of deep intronic regions by the last, which leads the sequencing away from exons (as does the WGS) but only for a few genes. Therefore, this panel would be in line with our proposal since it could be also considered a genomic approach (rather than an exomic one). Following the Reviewer’s advice, we have created two different columns in the Table, one for variants identified previously (exomic studies) and one for the variants detected thanks to our multi-omics approach.

97-108: section 2.2 seems to suggest that 4 of the 8 novel variants reported in this MS were discovered via the technologies that were highlighted as being inadequate in the Introduction, without any resort to WGS. In fact, it seems from lines 265-267 that WGS was required only in P3 and P5.

Of the 4 variants reported in section 2.2, 2 (the ones identified in GALE and DBT) were already reported in the literature and correspond to prior findings identified in exomic studies. The variant identified in IDUA is a novel change affecting splicing and it could be detected with this methodology because it is located in a +2 position, covered by exome studies. The last change (the one identified in OTC in P6) is also novel and it was detected thanks to a genomic metabolic panel customized in our lab. As stated in the paragraph above, this panel includes the intronic region of certain genes (OTC among them), therefore being considered as an additional omic (since c.541-277A>G could have never been identified by exomic studies).

Regarding the use of WGS in P3 and P5 that is completely true. But given that our work focuses on the use of multiple omics (WGS, RNA-Seq, functional genomics…) others of these techniques were needed to complete the diagnosis of the rest of the patients.

We share the Reviewer’s concern regarding the difficulty of understanding these results. To address this issue, we have rearranged the structure of section 2.2 and we have also changed the heading of this and the following sections.

103-104 & 162: can we please have a reference for each of the previously described variants? Also, I think it would be helpful to compare the strategy with which they were previously detected with the detection strategy in this study.

Following the Reviewer’s recommendation, we have included the references in which the previously described variants were reported (lines 103, 107 and 167).

244: the wording here, namely “leaving many patients without genetic confirmation of their condition” reinforces my feedback on lines 49-52. The problem is lack of diagnosis, not misdiagnosis.

We share the same concern as the Reviewer, agreeing that the problem is a lack of diagnosis and that our terminology was wrong. As mentioned in the comment on lines 49-52, we have changed the term “misdiagnosis” to “lack of diagnosis” since the former was not being used properly.

245-247: I am not sufficiently cognate to know of previous relevant literature, but I would be most surprised if no-one has ever suggested such a strategy before. In other words, are the authors certain that it is valid for them to imply that they are the first to suggest this strategy?

Thanks to the Reviewer’s correction we have realized that we forgot to add the reference to other work suggesting this strategy (line 258).

Reviewer 2 Report

ijms 1945679 v.1; Soriano-Sexto et al

The availability of automated methods for DNA sequencing, transcription and functional analyses has, & is having, a radical effect on our conception of the mechanisms by which DNA mutations result in inherited metabolic disease. We are still learning. The Authors applied a combination of genomic analyses to find the causative mutations in six individuals who had clinical and biochemical evidence of an inherited defect, but no previously identified mutation. Significantly they found that most of the mutations were in non-coding areas of DNA (four were deeply intronic), not detectable by exome sequencing. Eight novel variants affected splicing (6) or gene expression (2). These are important findings which extend our knowledge, and which will impact on clinical investigation. The studies were very thorough, carefully conducted and well-documented, although the manuscript was not an easy read. Despite the falling costs, these investigations are still very expensive and demand considerable interpretive expertise. Hence their use must be restricted. I think this should be highlighted briefly in the Discussion, and circumstances in which the tests are warranted, enumerated. Also needing emphasis is the crucial role of careful clinical observation and documentation and use of appropriate biochemical tests in the initial work-up. These are essential diagnostic ‘clues’ for directing DNA testing.

About the study:

i) Was it the case that none had had mutation analyses for their suspected disorder prior to this study?, or had they had tests yielding negative results? If they had, it would add impact to add this information to Table 1.

ii) What prompted the new genetic investigations? Were they for counselling or pedigree testing? Or to assess the merits of combined analyses as a research study?

iii) Table 1 P1. Was galactose-1-phosphate uridylyl transferase not measured at presentation? This is an essential emergency test for presumptive galactosemia.

iv) Table 1 P2. Surprising that dermatan sulfate was not increased in the urine considering iduronidase activity 0%

v) Table 1 P4. In view of his on-going gait problems, had he had mutation analyses with negative findings before the studies aged 28 years?

vi) I wonder whether it might be easier to follow if Table 1 was split so that this Table was confined to the working clinical diagnoses, and the observations & test results (including enzyme assays) on which these were based. The definitive gene abnormalities could then be presented in greater depth in a separate Table, later in the text. Suggested headings for this Table might be: mutated gene; age of definitive diagnosis; allele 1 (pat): mutation; site & consequence (eg deep intronic, splice site, inserted pseudo exon), informative analyses (coded- eg 2 [RNaseq], 3 [WGS], 5 [minigene—etc); allele 2 (mat) similarly.

You could then point out how the combined testing had provided a diagnosis when one was lacking previously (eg P4 ‘hyperphenylalaninemia’ now PTS deficiency), refined the diagnosis significantly (eg P3 ‘maple syrup urine disease to DBT (E2) deficiency), and/or provided precise details of the mutations which would facilitate prenatal screening or family counselling.

Odd points

vii) abstract L 28: NGS confirmation is now proposed for confirmation of most newly diagnosed IEM-not just those from neonatal screens.

viii) Para 2.2 L108: 'GT site' in full (a ref would be useful)

ix) Para 2.3 L113: 'AEI' in full -presumably allelic expression imbalance?

x) Fig 1 (d) L 142-Ref needed for Sashimi plot

Author Response

The availability of automated methods for DNA sequencing, transcription and functional analyses has, & is having, a radical effect on our conception of the mechanisms by which DNA mutations result in inherited metabolic disease. We are still learning. The Authors applied a combination of genomic analyses to find the causative mutations in six individuals who had clinical and biochemical evidence of an inherited defect, but no previously identified mutation. Significantly they found that most of the mutations were in non-coding areas of DNA (four were deeply intronic), not detectable by exome sequencing. Eight novel variants affected splicing (6) or gene expression (2). These are important findings which extend our knowledge, and which will impact on clinical investigation. The studies were very thorough, carefully conducted and well-documented, although the manuscript was not an easy read. Despite the falling costs, these investigations are still very expensive and demand considerable interpretive expertise. Hence their use must be restricted. I think this should be highlighted briefly in the Discussion, and circumstances in which the tests are warranted, enumerated. Also needing emphasis is the crucial role of careful clinical observation and documentation and use of appropriate biochemical tests in the initial work-up. These are essential diagnostic ‘clues’ for directing DNA testing.

We thank Reviewer #2 for the careful revision of the manuscript and the encouraging comments regarding our investigation.

About the study:

i) Was it the case that none had had mutation analyses for their suspected disorder prior to this study? or had they had tests yielding negative results? If they had, it would add impact to add this information to Table 1.

All patients had a first exomic analysis when the disorder was suspected clinical/biochemically. In all cases, this study yielded negative results since they did not identify any candidate variants or only one in autosomal recessive disorders. Following the Reviewer’s annotations, we have specified it in more detail in the text in section 2.2 (lines 100-110) and have also emphasized it in the Table 2.

ii) What prompted the new genetic investigations? Were they for counselling or pedigree testing? Or to assess the merits of combined analyses as a research study?

The additional analyses presented in this work were prompted for appropriate genetic counseling and preventive medicine (i.e prenatal diagnosis or preimplantation approach). In some cases, a tailored treatment according to the affected gene or even mutation-specific therapies could be applied. To clarify this issue we have included the sentence “Genetic confirmation, even in IEM, is essential for appropriated genetic counseling and in some cases for the application of gene or mutation-specific therapies.” In the Discussion (lines 266-268).

iii) Table 1 P1. Was galactose-1-phosphate uridylyl transferase not measured at presentation? This is an essential emergency test for presumptive galactosemia.

Thank you very much for this comment. The diagnosis protocol when we have a suspicion of galactosemia includes the enzymatic analysis. We performed, all three enzymatic activities (GALT, GALK and GALE). In the submitted MS we only indicated the affected enzymatic activity but we agree that all the results are informative. Thanks to the Reviewer’s advice we have also included this information in Table 1 in addition to other no-affected biochemical findings in all patients.

iv) Table 1 P2. Surprising that dermatan sulfate was not increased in the urine considering iduronidase activity 0%

Dermatan sulfate was shown slightly increased in P2 at diagnosis. We agree that all biochemical findings should be included to better understand the genotype/phenotype correlation. Following the annotation made by the Reviewer we have added this information to Table 1.

v) Table 1 P4. In view of his on-going gait problems, had he had mutation analyses with negative findings before the studies aged 28 years?

P3 had mutation analyses carried out since 1 year of age (time at biochemical/clinical diagnosis). These studies could only detect variant c.827T>G, remaining undiagnosed until aged 28 years when the second variant was identified, therefore, achieving a complete diagnosis. In this case, we analyzed several years ago the DBT mRNA and never detected an aberrant transcript. The advent in the clinical lab of RNA-Seq and genomic analysis has enabled us to solve this case several years later.

vi) I wonder whether it might be easier to follow if Table 1 was split so that this Table was confined to the working clinical diagnoses, and the observations & test results (including enzyme assays) on which these were based. The definitive gene abnormalities could then be presented in greater depth in a separate Table, later in the text. Suggested headings for this Table might be: mutated gene; age of definitive diagnosis; allele 1 (pat): mutation; site & consequence (eg deep intronic, splice site, inserted pseudo exon), informative analyses (coded- eg 2 [RNaseq], 3 [WGS], 5 [minigene—etc); allele 2 (mat) similarly.

You could then point out how the combined testing had provided a diagnosis when one was lacking previously (eg P4 ‘hyperphenylalaninemia’ now PTS deficiency), refined the diagnosis significantly (eg P3 ‘maple syrup urine disease to DBT (E2) deficiency), and/or provided precise details of the mutations which would facilitate prenatal screening or family counselling.

We really thank the Reviewer for this comment. We have split Table 1 in two different tables to try to make all the information clearer.  

Odd points

vii) abstract L 28: NGS confirmation is now proposed for confirmation of most newly diagnosed IEM-not just those from neonatal screens.

Following the recommendation made by the Reviewer, we have explained this further “Next generation sequencing has transformed the diagnosis of IEM, leading to its proposed use as a second-tier technology for confirming cases detected by clinical/biochemical analysis, i.e. newborn screening.”

viii) Para 2.2 L108: 'GT site' in full (a ref would be useful)

We share our concern with the Reviewer that a reference would be useful in this regard. We have added it (line 177).

ix) Para 2.3 L113: 'AEI' in full -presumably allelic expression imbalance?

Full explanation of AEI (allelic expression imbalance) was already described in the introduction (lines 73 and 74).

x) Fig 1 (d) L 142-Ref needed for Sashimi plot

We have followed the Reviewer’s advice and added the way the Sashimi plots were designed.
